# mLongT5: A Multilingual and Efficient Text-To-Text Transformer for Longer Sequences

**David Uthus, Santiago Ontañón, Joshua Ainslie, Mandy Guo**
Google Research
{duthus, santiontanon, jainslie, xyguo}@google.com

## Abstract

We present our work on developing a multilingual, efficient text-to-text transformer that is suitable for handling long inputs. This model, called mLongT5, builds upon the architecture of LongT5, while leveraging the multilingual datasets used for pretraining mT5 and the pretraining tasks of UL2. We evaluate this model on a variety of multilingual summarization and question-answering tasks, and the results show stronger performance for mLongT5 when compared to existing multilingual models such as mBART or M-BERT.

## 1 Introduction

In recent years, there has been development of making transformer-based models more efficient so that they can handle longer input sequences. Many of the models though have been English-only, making them inapplicable to other languages.

In this paper, we present our work in extending one of these models to be able to handle multilingual data. Our model, called *mLongT5*, takes advantage of the efficient architecture of LongT5 (Guo et al., 2022), and has been pretrained on the multilingual mC4 dataset (Xue et al., 2021) to be able to work on multilingual tasks. We have applied mLongT5 to a variety of multilingual summarization and question-answering tasks, and results show that mLongT5 exhibits strong performance in these domains.

The configurations[1] and checkpoints[2] have all been open-sourced.

## 2 Related Work

There are two areas of related work – efficient transformer models that can handle long inputs, and multilingual models.

---

[1] https://github.com/google/flaxformer/tree/main/flaxformer/t5x/configs/longt5/models
[2] https://github.com/google-research/longt5

There has been much interest of late in making transformer models more efficient, such as to handle longer inputs. Example of these include ETC (Ainslie et al., 2020), Big Bird (Zaheer et al., 2020), LongT5 (Guo et al., 2022), and Longformer (Beltagy et al., 2020). These models were successful in taking various approaches to address the quadratic growth of the attention mechanism in transformers. Unfortunately though, these models are trained on English datasets, limiting their use in multilingual domains.

With respect to multilingual models, these would include mT5 (Xue et al., 2021), mBART (Liu et al., 2020), and the recent umT5 (Chung et al., 2023). These models re-used architectures used by English models but are pretrained on a larger, multilingual corpus, with mT5 and umT5 trained on 101 languages and mBART on 25. While these models showed strong performance on being able to handle a wide variety of languages, they suffered the same restrictions as their original English models on not being able to scale up to longer sequences.

## 3 Model

mLongT5 builds upon the architecture of LongT5 (Guo et al., 2022). LongT5 was developed to efficiently handle long inputs by utilizing a more efficient attention mechanism. The model was shown to have strong performance on a variety of downstream tasks, and thus is the foundation for mLongT5.

### 3.1 Datasets

To make mLongT5 multilingual, we leverage the mC4 dataset used for training the multilingual model mT5 (Xue et al., 2021), which consists of 101 languages. This dataset has recently been updated, as described by Chung et al. (2023), and was used for training umT5 and creating a new SentencePiece model (Kudo and Richardson, 2018). As such, we then make use of the same SentencePiece

model used for umT5, thus allowing mLongT5 to handle multilingual inputs.

## 3.2 Pretraining Tasks

One key difference with our model and LongT5 is the changing of tasks for pretraining the model. LongT5 made use of PEGASUS' Principle Sentences Generation (PSG) (Zhang et al., 2020) for pretraining its models. While this was shown to have strong performance for various downstream tasks, the one weakness of PSG is that it is less suitable for multilingual training. PSG relies on being able to split a piece of text into sentences, with current implementation best suited for Latin-based languages. The need to break text into sentences properly for 101 different languages makes it then a challenging task to use in a multilingual setting.

To overcome this, we instead decided to apply UL2's pretraining tasks (Tay et al., 2022). Their pretraining task, called Mixture-of-Denoisers (MoD), has the model learning from a mixture of tasks, and has been shown to work better than T5's original pretraining task (Raffel et al., 2019). More importantly, MoD can be more easily applied to other languages compared to PSG, thus making it ideal for pretraining mLongT5.

## 3.3 Pretraining Details

Pretraining mLongT5 has many similarities to how LongT5 was pretrained. It is pretrained for one million steps, and we pretrained model sizes of Base, Large, and XL. We also use the same pretraining lengths, 4,096 for the inputs and 910 for the targets. One small difference is increasing the batch size from 128 to 256, allowing the model to train on the same number of tokens as mT5. For the mC4 dataset, we used version 3.1.0, which is the version update by Chung et al. (2023). For dataset sampling, we use the UniMax sampling method (Chung et al., 2023).

Instead of PSG as pretraining task, we apply MoD, using the same configuration as defined in the original UL2 task definition. The only exception is that we do not use 0.5 corruption rate (using only corruption rate of 0.15), as our input lengths (4096) are much longer than our target lengths (910), making a corruption rate of 0.5 unfeasible.

All models were pretrained using 256 TPUv4 chips. Wall time to pretrain these models was 1.9 days for Base, 3.7 days for Large, and 12.4 days for XL.

# 4 Results

As with the original LongT5 paper, we look at two domains for evaluating our model: summarization and question answering.

For all of these tasks, we use the default values as used for T5 finetuning, only explicitly setting the input and target lengths as described in the tasks below.

## 4.1 Summarization

The three summarization tasks we are looking at are:

- MLSUM (Scialom et al., 2020): a collection of newspaper articles and their corresponding summaries in five languages: French, German, Spanish, Russian, and Turkish.

- XL-Sum (Hasan et al., 2021): a collection of BBC articles and summaries in 44 languages.

- WikiLingua (Ladhak et al., 2020): a collection of documents from WikiHow (in Spanish, Turkish, Russian, and Vietnamese) that have been translated and summarized into English. For this task, we are using the GEM (Gehrmann et al., 2021) version of the datasets, allowing us to make use of their fixes in the splitting of the datasets for training and testing.

These tasks allow us to explore summarization where the task involves documents and their summaries in the same language (MLSUM, XL-Sum), or where the task involves both translation and summarization at the same time (WikiLingua).

We note that with respect to task lengths, these multilingual tasks are not very long when compared to the tasks covered in the original LongT5 paper. There is unfortunately a lack of lengthy, multilingual summarization tasks available, thus we use these three for comparisons. As such, we tested with input lengths of 4k for input and 512 for output, which covers most documents for all the above tasks.

For all these tasks, we report standard ROUGE scores (ROUGE-1, ROUGE-2, and ROUGE-L).

### 4.1.1 MLSUM

Table 1 shows our results for the MLSUM task. We are comparing to the M-BERT (Devlin, 2018) model used in the original paper. The authors only

| | **FR** | | |
|---|---|---|---|
| **Approach** | R-1 | R-2 | R-L |
| M-BERT | - | - | 25.09 |
| mLongT5 (base) | 30.79 | 14.16 | 23.83 |
| mLongT5 (large) | 31.44 | 14.74 | 24.36 |
| mLongT5 (xl) | 32.18 | 15.68 | **25.18** |
| | **DE** | | |
| **Approach** | R-1 | R-2 | R-L |
| M-BERT | - | - | 42.01 |
| mLongT5 (base) | 45.60 | 35.31 | 42.22 |
| mLongT5 (large) | 46.21 | 35.68 | 42.71 |
| mLongT5 (xl) | 46.95 | 36.36 | **43.45** |
| | **ES** | | |
| **Approach** | R-1 | R-2 | R-L |
| M-BERT | - | - | 20.44 |
| mLongT5 (base) | 28.78 | 10.98 | 23.15 |
| mLongT5 (large) | 29.05 | 11.58 | 23.50 |
| mLongT5 (xl) | 30.36 | 12.77 | **24.73** |
| | **TR** | | |
| **Approach** | R-1 | R-2 | R-L |
| M-BERT | - | - | 32.94 |
| mLongT5 (base) | 44.18 | 30.86 | 38.60 |
| mLongT5 (large) | 44.92 | 31.55 | 39.29 |
| mLongT5 (xl) | 45.73 | 32.80 | **40.26** |
| | **RU** | | |
| **Approach** | R-1 | R-2 | R-L |
| M-BERT | - | - | **9.48** |
| mLongT5 (base) | 7.73 | 1.78 | 7.22 |
| mLongT5 (large) | 7.71 | 1.86 | 7.23 |
| mLongT5 (xl) | 8.85 | 2.67 | 8.42 |

Table 1: MLSUM results comparing mLongT5 with the original model M-BERT. Note that the original paper only reported ROUGE-L scores, while we also report ROUGE-1 and ROUGE-2.

reported ROUGE-L scores, while we also report ROUGE-1 and ROUGE-2 scores.

Looking at the ROUGE-L scores, we can see that mLongT5 performs comparably to M-BERT for French, while doing better than M-BERT for all model sizes in German, Spanish, and Turkish. It is only with Russian does it do slightly worse. As noted in the original paper, Russian was the hardest language for language models, due to having a much smaller dataset when compared to the other languages in the corpus and a higher rate of novelty (words found in the summary but not in the input document). Additionally, as we mentioned before, the dataset input lengths are not very long, thus models with full attention can take better advantage of the short lengths compared to mLongT5. This can then contribute to mLongT5 not performing as well for this instance.

### 4.1.2 XL-Sum

For XL-Sum, we finetuned the model in a similar approach to the original paper – we finetuned on a mixture of all the languages for 50,000 steps, and then performed tests for each of the individual languages from this single model.

Table 2 shows a subset of the languages (the full results can be seen in Appendix A). We highlight languages that had longer input lengths (due to both the length of the original documents and how they are then subsequently tokenized by the SPM).

As we can see, mLongT5 performed well compared to mT5 for these lengthier inputs. When comparing base to base, it did slightly worse, as expected with mT5 having full attention. The original LongT5 model, when finetuned on datasets that are of shorter lengths, had also shown slightly worse performance when compared to a model of full attention. We are seeing similar results here. But mLongT5 is able to more easily scale to larger model sizes, and as such, we can see stronger results as we increase the size of the model.

### 4.1.3 WikiLingua

The final summarization task is WikiLingua, with results shown in Table 3. This task requires both translation and summarization, with the task translating from a full document of another language into an English summary. As previously mentioned we are using the GEM version of this task, and compare our results to the mT5 model on their leaderboard.

As shown in the results, mLongT5 tends to do better for many of the model sizes across the 4 languages, with only slightly worse performance with XL size for Spanish.

### 4.2 Question-Answering

For question-answering, we applied mLongT5 to TyDi QA (Clark et al., 2020). TyDi QA is a multilingual task covering 11 languages, trying to answer questions given a Wikipedia article. There are two versions of this task, and we focus on the Minimal Answer Span Task, in which one is trying to either find the minimal span that answer the question, give a yes/no answer if the question is a yes/no question, or Null if the question cannot be answered given the article.

Similar to the original LongT5 paper and their application to Natural Questions, we have redefined this task from extracting answer spans to a seq2seq task of generating answer texts. The

| Language | mT5 (base) | | | mLongT5 (base) | | | mLongT5 (large) | | | mLongT5 (xl) | | |
|---|---|---|---|---|---|---|---|---|---|---|---|---|
| | R-1 | R-2 | R-L | R-1 | R-2 | R-L | R-1 | R-2 | R-L | R-1 | R-2 | R-L |
| Gujarati | 21.96 | 7.74 | 19.86 | 19.59 | 6.08 | 17.61 | 22.38 | 7.94 | 20.15 | **25.52** | **9.92** | **22.78** |
| Marathi | 22.01 | 9.54 | 19.92 | 20.33 | 8.62 | 18.41 | 23.35 | 10.56 | 21.22 | **25.90** | **12.03** | **23.07** |
| Punjabi | 30.70 | 12.21 | 25.52 | 28.61 | 10.43 | 23.66 | 31.92 | 12.75 | 26.17 | **34.45** | **14.81** | **28.42** |
| Serbian (Cyrillic) | 23.78 | 7.98 | 20.14 | 20.30 | 5.86 | 16.74 | 21.92 | 6.98 | 18.35 | **27.51** | **11.46** | **23.49** |
| Serbian (Latin) | 21.64 | 6.66 | 18.23 | 18.14 | 4.75 | 14.96 | 21.79 | 6.92 | 18.14 | **25.86** | **10.17** | **21.76** |
| Vietnamese | 32.88 | 16.22 | 26.08 | 31.58 | 15.41 | 25.02 | 34.54 | 17.63 | 27.59 | **38.17** | **20.49** | **30.98** |

Table 2: Results for XL-Sum, focusing on languages that have lengthier inputs. The rest of the results can be seen in the Appendix A.

| Approach | ES-EN | | | TR-EN | | | RU-EN | | | VI-EN | | |
|---|---|---|---|---|---|---|---|---|---|---|---|---|
| | R-1 | R-2 | R-L | R-1 | R-2 | R-L | R-1 | R-2 | R-L | R-1 | R-2 | R-L |
| mT5 (base) | 30.9 | 10.6 | 26.4 | 32.0 | 13.1 | 26.0 | 27.3 | 8.6 | 23.3 | 25.6 | 7.7 | 21.5 |
| mT5 (large) | 34.2 | 12.6 | 29.1 | 34.0 | 14.5 | 27.5 | 32.3 | 11.2 | 26.9 | 32.1 | 10.9 | 26.0 |
| mT5 (xl) | **41.2** | 17.2 | **34.6** | 40.0 | 18.3 | 33.3 | 37.2 | 14.6 | 30.9 | 37.6 | 14.9 | 31.2 |
| mLongT5 (base) | 36.1 | 14.0 | 30.3 | 34.5 | 14.9 | 28.6 | 32.4 | 11.6 | 26.5 | 32.3 | 11.7 | 26.4 |
| mLongT5 (large) | 38.2 | 15.5 | 32.0 | 38.1 | 17.5 | 32.0 | 34.4 | 13.1 | 28.5 | 35.1 | 13.8 | 29.1 |
| mLongT5 (xl) | 40.8 | **17.6** | 34.3 | **42.5** | **20.9** | **36.7** | **37.6** | **15.7** | **31.8** | **38.7** | **16.6** | **32.8** |

Table 3: WikiLingua summarization results. These results are using the GEM version of the task.

results shown will then differ from the TyDi QA leaderboard. As such, we have also run the similar mT5 model on the same task to get a baseline to compare against. Additionally, as the test set is not available for this task, we use 90% of the training data as the train set and remaining 10% as the dev set, and use the original dev set as our test set for reporting metrics.

Unlike the summarization tasks, TyDi QA has much longer input lengths – mean of 5,148 tokens and $90^{th}$ percentile of 12,967 tokens when tokenized with the SentencePiece model. As such, for mT5 we tested with input lengths between 512 and 4k, while for mLongT5 we tested with input lengths between 4k and 16k.

Table 4 show the results of running mT5 and mLongT5 on this dataset. For this task, we report metrics of Exact Match (EM) and F1 score. As can be seen in the results, mLongT5 is able to better answer the questions given that it can handle longer input sequences.

## 5 Conclusion

We have presented our new model mLongT5. It has the benefits of the efficient architecture of LongT5, with the ability to handle multingual inputs and outputs. As our report shows, the model is able to perform well on a variety of summarization and question-answering tasks.

| Approach | EM | F1 |
|---|---|---|
| mT5 (base - 512 input) | 37.16 | 49.99 |
| mT5 (base - 1k input) | 43.09 | 56.36 |
| mT5 (base - 2k input) | 44.63 | 58.12 |
| mT5 (base - 4k input) | 45.41 | 58.63 |
| mT5 (large - 512 input) | 40.96 | 54.08 |
| mT5 (large - 4k input) | 52.77 | 66.54 |
| mT5 (xl - 512 input) | 43.84 | 56.98 |
| mT5 (xl - 4k input) | 55.03 | 68.26 |
| mLongT5 (base - 4k input) | 50.76 | 62.74 |
| mLongT5 (base - 8k input) | 51.21 | 63.66 |
| mLongT5 (base - 16k input) | 52.43 | 64.51 |
| mLongT5 (large - 4k input) | 54.04 | 66.75 |
| mLongT5 (large - 8k input) | 55.56 | 68.26 |
| mLongT5 (large - 16k input) | 55.93 | 68.66 |
| mLongT5 (xl - 4k input) | 58.52 | 70.86 |
| mLongT5 (xl - 8k input) | 59.6 | 71.86 |
| mLongT5 (xl - 16k input) | **60.42** | **72.63** |

Table 4: TyDi QA results.

## Limitations

mLongT5 has the same limitations as seen in the original LongT5 model, in that they are more suited for tasks of lengthier inputs. Tasks with shorter inputs will be better served by models like mT5 and umT5, which can take advantage of full attention.

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

# A    XL-Sum

We show the full results of running our mLongT5 models on XL-Sum in Table 5. These results are those that had been uploaded to GitHub [3] by the authors along with the updated datasets.

When computing ROUGE scores, we use similar computations as done in the respective paper, with exceptions to Chinese, Japanese and Thai. For these languages, we use the SPM we used in our model for the tokenization of the results in order to compute ROUGE.

---

[3]https://github.com/csebuetnlp/xl-sum

| Language | mT5 (base) | | | mLongT5 (base) | | | mLongT5 (large) | | | mLongT5 (xl) | | |
|---|---|---|---|---|---|---|---|---|---|---|---|---|
| | R-1 | R-2 | R-L | R-1 | R-2 | R-L | R-1 | R-2 | R-L | R-1 | R-2 | R-L |
| Amharic | 20.05 | 7.41 | 18.08 | 16.70 | 5.91 | 14.73 | 20.29 | 7.99 | 18.09 | 22.37 | 8.90 | 19.91 |
| Arabic | 34.91 | 14.79 | 29.16 | 26.39 | 11.01 | 22.45 | 27.65 | 12.25 | 23.57 | 32.09 | 15.04 | 27.74 |
| Azerbaijani | 21.42 | 9.52 | 19.33 | 17.52 | 7.10 | 15.77 | 19.92 | 8.80 | 18.08 | 22.68 | 9.89 | 20.36 |
| Bengali | 29.57 | 12.11 | 25.13 | 21.39 | 8.22 | 18.65 | 24.69 | 10.04 | 21.25 | 26.83 | 11.32 | 22.86 |
| Burmese | 15.96 | 5.15 | 14.18 | 45.28 | 26.62 | 34.76 | 49.07 | 29.52 | 38.10 | 51.60 | 31.69 | 40.20 |
| Chinese (Simp.) | 39.41 | 17.79 | 33.41 | 38.90 | 21.78 | 32.59 | 42.62 | 24.70 | 35.80 | 48.42 | 29.99 | 41.28 |
| Chinese (Trad.) | 37.19 | 17.14 | 31.62 | 39.45 | 22.40 | 32.51 | 43.32 | 25.56 | 35.95 | 48.82 | 30.80 | 41.18 |
| English | 37.60 | 15.15 | 29.88 | 32.85 | 11.38 | 25.64 | 35.59 | 13.63 | 28.02 | 39.51 | 17.00 | 31.77 |
| French | 35.34 | 16.17 | 28.20 | 30.06 | 12.93 | 24.21 | 31.88 | 14.32 | 25.61 | 34.82 | 16.17 | 28.11 |
| Gujarati | 21.96 | 7.74 | 19.86 | 19.59 | 6.08 | 17.61 | 22.38 | 7.94 | 20.15 | 25.52 | 9.92 | 22.78 |
| Hausa | 39.44 | 17.68 | 31.67 | 34.61 | 13.73 | 27.30 | 38.04 | 16.07 | 30.32 | 40.58 | 18.57 | 32.52 |
| Hindi | 38.59 | 16.88 | 32.01 | 34.81 | 14.29 | 28.71 | 37.42 | 16.71 | 31.22 | 40.92 | 19.73 | 34.41 |
| Igbo | 31.61 | 10.16 | 24.53 | 25.82 | 8.05 | 20.19 | 30.41 | 10.01 | 23.68 | 31.31 | 9.88 | 24.07 |
| Indonesian | 37.00 | 17.02 | 30.76 | 32.15 | 13.05 | 26.59 | 35.17 | 15.23 | 29.07 | 38.87 | 18.00 | 32.64 |
| Japanese | 48.15 | 23.85 | 37.36 | 45.56 | 27.12 | 36.51 | 48.60 | 29.95 | 39.00 | 50.77 | 32.06 | 40.79 |
| Kirundi | 31.99 | 14.37 | 25.83 | 25.61 | 10.07 | 20.26 | 29.36 | 12.78 | 23.67 | 31.67 | 14.55 | 25.50 |
| Korean | 23.67 | 11.45 | 22.36 | 20.25 | 9.20 | 19.00 | 23.18 | 10.42 | 21.38 | 25.30 | 11.63 | 23.31 |
| Kyrgyz | 18.38 | 7.96 | 16.50 | 14.08 | 5.27 | 12.46 | 16.01 | 6.30 | 14.14 | 18.19 | 7.81 | 16.00 |
| Marathi | 22.01 | 9.54 | 19.92 | 20.33 | 8.62 | 18.41 | 23.35 | 10.56 | 21.22 | 25.90 | 12.03 | 23.07 |
| Nepali | 26.65 | 10.25 | 24.28 | 23.96 | 8.94 | 21.80 | 26.24 | 10.33 | 23.91 | 28.87 | 11.59 | 26.17 |
| Oromo | 18.70 | 6.17 | 16.19 | 14.88 | 4.38 | 12.71 | 17.91 | 5.65 | 15.28 | 19.52 | 6.50 | 17.18 |
| Pashto | 38.47 | 15.55 | 31.91 | 35.01 | 13.79 | 28.84 | 38.63 | 16.06 | 32.00 | 41.37 | 17.61 | 33.92 |
| Persian | 36.94 | 16.19 | 30.07 | 35.47 | 14.66 | 28.40 | 37.70 | 16.45 | 30.49 | 40.64 | 18.89 | 33.16 |
| Pidgin | 37.96 | 15.12 | 29.87 | 33.86 | 12.01 | 26.68 | 35.86 | 13.72 | 28.24 | 38.01 | 15.08 | 29.78 |
| Portuguese | 37.17 | 15.90 | 28.56 | 31.67 | 12.51 | 24.46 | 34.04 | 14.51 | 26.65 | 37.66 | 17.57 | 29.88 |
| Punjabi | 30.70 | 12.21 | 25.52 | 28.61 | 10.43 | 23.66 | 31.92 | 12.75 | 26.17 | 34.45 | 14.81 | 28.42 |
| Russian | 32.22 | 13.64 | 26.17 | 22.11 | 8.29 | 18.62 | 24.39 | 10.00 | 20.54 | 28.20 | 12.72 | 23.91 |
| Scottish Gaelic | 29.02 | 10.99 | 22.88 | 26.98 | 8.87 | 21.57 | 29.80 | 10.64 | 23.44 | 31.74 | 12.61 | 25.65 |
| Serbian (Cyrillic) | 23.78 | 7.98 | 20.14 | 20.30 | 5.86 | 16.74 | 21.92 | 6.98 | 18.35 | 27.51 | 11.46 | 23.49 |
| Serbian (Latin) | 21.64 | 6.66 | 18.23 | 18.14 | 4.75 | 14.96 | 21.79 | 6.92 | 18.14 | 25.86 | 10.17 | 21.76 |
| Sinhala | 27.29 | 13.38 | 23.47 | 22.69 | 10.02 | 19.96 | 25.24 | 11.52 | 21.98 | 27.78 | 13.20 | 24.45 |
| Somali | 31.56 | 11.58 | 24.22 | 27.85 | 9.08 | 21.10 | 30.29 | 10.69 | 23.29 | 31.64 | 11.11 | 24.28 |
| Spanish | 31.51 | 11.88 | 24.07 | 26.82 | 9.05 | 20.47 | 28.71 | 10.56 | 22.04 | 32.20 | 13.10 | 24.88 |
| Swahili | 37.67 | 17.85 | 30.91 | 31.79 | 13.25 | 25.67 | 34.29 | 15.22 | 27.82 | 37.29 | 17.22 | 30.96 |
| Tamil | 24.33 | 11.06 | 22.07 | 20.68 | 8.67 | 18.71 | 24.08 | 10.74 | 21.71 | 26.81 | 12.23 | 24.21 |
| Telugu | 19.86 | 7.03 | 17.61 | 15.11 | 4.69 | 13.48 | 17.98 | 6.12 | 16.10 | 21.20 | 7.77 | 18.88 |
| Thai | 37.40 | 17.28 | 28.88 | 35.98 | 21.39 | 26.65 | 38.11 | 22.92 | 28.26 | 40.70 | 25.23 | 30.12 |
| Tigrinya | 25.32 | 8.02 | 21.17 | 22.27 | 7.08 | 18.61 | 26.30 | 8.90 | 22.05 | 28.53 | 10.13 | 24.05 |
| Turkish | 32.93 | 15.57 | 29.26 | 25.52 | 11.54 | 22.83 | 28.56 | 13.62 | 25.72 | 31.33 | 15.61 | 28.20 |
| Ukrainian | 23.99 | 10.14 | 20.92 | 20.97 | 8.16 | 18.17 | 23.34 | 9.74 | 20.29 | 27.05 | 12.16 | 23.68 |
| Urdu | 39.56 | 18.37 | 32.84 | 37.11 | 15.97 | 30.14 | 39.90 | 18.53 | 32.75 | 43.03 | 21.40 | 35.72 |
| Uzbek | 16.83 | 6.34 | 15.41 | 14.60 | 5.36 | 13.39 | 17.26 | 6.42 | 15.49 | 19.18 | 7.80 | 17.29 |
| Vietnamese | 32.88 | 16.22 | 26.08 | 31.58 | 15.41 | 25.02 | 34.54 | 17.63 | 27.59 | 38.17 | 20.49 | 30.98 |
| Welsh | 32.66 | 11.60 | 26.12 | 29.96 | 9.40 | 23.96 | 33.66 | 12.26 | 27.01 | 36.49 | 15.34 | 29.79 |
| Yoruba | 31.66 | 11.66 | 25.09 | 25.87 | 8.99 | 20.27 | 29.49 | 10.50 | 23.26 | 32.20 | 12.34 | 25.84 |

Table 5: Full results for XL-Sum.