# OpenReview forum: "mLongT5: A Multilingual and Efficient Text-To-Text Transformer for Longer Sequences"
_EMNLP/2023/Conference — EMNLP 2023 Findings_

### Official Review · Reviewer_bAoP · 2023-07-31

**Soundness:** 4

**Excitement:**

3: Ambivalent: It has merits (e.g., it reports state-of-the-art results, the idea is nice), but there are key weaknesses (e.g., it describes incremental work), and it can significantly benefit from another round of revision. However, I won't object to accepting it if my co-reviewers champion it.

**Paper Topic And Main Contributions:**

The paper reports the procedure for training mLongT5, a transformer model to handle long input sequences with an extensive evaluation of its abilities across a number of languages. The main contribution is in making a publicly available pre-trained model.

**Reasons To Accept:**

  1. Practically useful multilingual language model which can handle long input sequences
  2. Extensive evaluation with respect to the number of languages and evaluation scenarios.

**Reasons To Reject:**

 1. Using Rouge for evaluation is not warranted. It has been shown that it is far less reliable with respect to evaluation of neural models as it relies on exact lexical matches. BertScore is more reliable and there is no inherent danger in using BertScore to evaluate mBERT as both use different mechanisms.
 2. There is little innovation in the pipeline, but having a reporting paper published in a peer-reviewed conference is better than merely an Arxiv publication.


**Reproducibility:**

4: Could mostly reproduce the results, but there may be some variation because of sample variance or minor variations in their interpretation of the protocol or method.

**Reviewer Confidence:**

4: Quite sure. I tried to check the important points carefully. It's unlikely, though conceivable, that I missed something that should affect my ratings.

---

> ### Author Rebuttal · Authors · 2023-08-28
>
> Thank you for your review and comments.
>
> For reporting ROUGE in our results, while we agree ROUGE is not the best metrics (compared to recent metrics like BARTScore, T5-ANLI, SMART, and RISE), ROUGE does make it easier to compare to past approaches as all these part approaches report ROUGE scores. Additionally, as mentioned in the BERTScore paper, multilingual BERT has been shown to not work as well for low-resource languages when doing scores, which would make it less reliable than a metric like ROUGE which tends to be language agnostic.
>
> If this was a summarization focused paper, we would report additional metrics, but as this is more showing the results of a general-purpose pre-trained model, we think using a standard metric as ROUGE would be enough for a reader to understand how our model compares to others. We will though add more explanation to the paper why we focused on using ROUGE for our metric.

---

### Official Review · Reviewer_5oDq · 2023-08-04

**Soundness:** 4

**Excitement:**

3: Ambivalent: It has merits (e.g., it reports state-of-the-art results, the idea is nice), but there are key weaknesses (e.g., it describes incremental work), and it can significantly benefit from another round of revision. However, I won't object to accepting it if my co-reviewers champion it.

**Paper Topic And Main Contributions:**

This paper presents a Multilingual LongT5 model (mLongT5), derived and extended from mT5, LongT5 and and UL2 by using a multilingual corpora. The mLongT5 model has been evaluated on  a variety of multilingual summarization and question-answering tasks, where the performance outperforms the existing multilingual models like mBERT and mT5.

**Questions For The Authors:**

1. In line 168 "As noted in the original paper, Russian was the hardest language for language models, due to having a much smaller dataset when compared to the other languages", cannot explain the results in a straight line. It seems like to display the unfairness between the datasets that mBERT and mLongT5 used. May you use the pre-training datasets that mBERT used to pre-train mLongT5 (or use the Russian part) to issue this problem?

2. In Table 2, mT5(base) outperforms mLongT5(base) on almost all languages. How you describe this phenomena?

3. All the experiments are less a discussion of computational cost. May you add a paragraph to compare the costs of pre-training for these models?

**Reasons To Accept:**

1. This paper integrates and extends the methods from mT5, Long T5 and UL2, and proposes several multilingual versions of LongT5 (base large xl), which achieve improvements on several multilingual downstream tasks.

2. The experiments are elaborated. Not only the model size is discussed, but also the input token length is compared (Table 4).

3. The code and datasets are open, which is good for the community.

4. Stable improvements are achieved during increasing the model size of mLongT5, as shown in the experiments, showing great potential and insights for larger language models.

**Reasons To Reject:**

1. Although the experiments are conducted well, there still lack some explanation for some experimental results. Such as, in Table 1, mLongT5(base) is not as good as mBERT(base) on FR evaluation; besides, as for the RU, the author just claims that Russian is the hardest language for language models for its sparsity. The explanation seems simple and even subjective. You could set a ablation study that mLongT5 is tasked to pre-trained on the same datasets that mBERT uses to issue this question.

2. As for the Table 4, the author compare the mT5 model and the mLongT5 model with the same token lengths to prove the statement that mLongT5 processes long text better. More datasets and existing models like XLM-Roberta should be discussed to extend the generalization.

**Reproducibility:**

5: Could easily reproduce the results.

**Reviewer Confidence:**

4: Quite sure. I tried to check the important points carefully. It's unlikely, though conceivable, that I missed something that should affect my ratings.

---

> ### Author Rebuttal · Authors · 2023-08-28
>
> Thank you for your review and comments.
>
> 1. It is possible that pretraining the models on the same corpus as mBERT was trained on could have led to closer/better results for Russian. Unfortunately though, pretraining the models on another dataset was not feasible with the computational resources we had available for this research project.
>
> 2. As mentioned earlier in the paper, the summarization datasets we used were on the smaller side in terms of input lengths (when compared to TyDi QA and the English summarization datasets used in original LongT5 paper), thus mLongT5 is not able to take as much of an advantage of its ability to handle longer inputs. As mentioned in the original LongT5 paper, for shorter inputs such as the CNN summarization dataset, having a model with full attention will perform better.
>
> 3. To understand your question, do you mean in terms of the amount of accelerators used for pretraining the 3 size models along with the amount of time taken? If so, then yes we will report our values in the paper.

---

### Official Review · Reviewer_NgeM · 2023-08-07

**Soundness:** 4

**Excitement:**

3: Ambivalent: It has merits (e.g., it reports state-of-the-art results, the idea is nice), but there are key weaknesses (e.g., it describes incremental work), and it can significantly benefit from another round of revision. However, I won't object to accepting it if my co-reviewers champion it.

**Paper Topic And Main Contributions:**

The paper proposes mLongT5, which extends LongT5 to the multilingual scenario. The paper provides detailed information of model pre-training, and conducts extensive experiments on MLSUM, XL-Sum, Wikilingua, and TyDi QA. The checkpoints will be open-sourced, which will facilitate the research on long text processing in NLP.

**Questions For The Authors:**

Would you please provide detailed hyperparameters of mLongT5 and mT5 for fine-tuning?

**Reasons To Accept:**

Training a multilingual LM that handles long input sequences is an important topic for multilingual research.

The open-sourced models will facilitate the research on long text processing in NLP, espcially for low-resource languages.

Extensive experiments on multiple datasets. The results in Table 4 shows that mLongT5 is good at handling long input sequences.

**Reasons To Reject:**

Directly extending LongT5 to the multilingual one is somewhat lacking in novelty.

**Reproducibility:**

3: Could reproduce the results with some difficulty. The settings of parameters are underspecified or subjectively determined; the training/evaluation data are not widely available.

**Reviewer Confidence:**

4: Quite sure. I tried to check the important points carefully. It's unlikely, though conceivable, that I missed something that should affect my ratings.

---

> ### Author Rebuttal · Authors · 2023-08-28
>
> Thank you for your review and comments.
>
> With respect to hyperparemeters for finetuning, we used the default values found in the open-sourced gin files for the respective models. The only value we explicitly set was batch size of 128. We will update the paper with this information.

---

### Meta-Review · Area_Chair_svH2 · 2023-09-19

**Recommendation:** 3

**Metareview:**

This paper presents multilingual LongT5 (mLongT5), a multilingual version of LongT5 which accepts longer context length than other Transformer-based models and reports experimental results in summarization and QA tasks. Overall, the reviewers found that the paper presents interesting observations and promising experimental results with mLongT5. However, a few points are raised by the reviewers such as Russian summarization results on the MLSUM dataset are worse than multilingual BERT without detailed analysis and using only ROUGE scores for summarization. Based on the reviews, the AC strongly recommends the authors to 1) add analysis on settings where the mLongT5 falls short when compared to mT5 or multilingual BERT, and 2) add at least one model-based evaluation metric since ROUGE scores are measuring only one aspect of generated summaries.

---

### Decision · Program_Chairs · 2023-10-07

**Decision:**

Accept-Findings

**Comment:**

This paper presents multilingual LongT5 (mLongT5), a multilingual version of LongT5 which accepts longer context length than other Transformer-based models and reports experimental results in summarization and QA tasks. Overall, the reviewers found that the paper presents interesting observations and promising experimental results with mLongT5. However, a few points are raised by the reviewers such as Russian summarization results on the MLSUM dataset are worse than multilingual BERT without detailed analysis and using only ROUGE scores for summarization. Based on the reviews, the AC strongly recommends the authors to 1) add analysis on settings where the mLongT5 falls short when compared to mT5 or multilingual BERT, and 2) add at least one model-based evaluation metric since ROUGE scores are measuring only one aspect of generated summaries.